# Fungal–Lactobacteria Consortia and Enzymatic Catalysis for Polylactic Acid Production

**DOI:** 10.3390/jof9030342

**Published:** 2023-03-10

**Authors:** Laura I. de Eugenio, Carlos Murguiondo, Sandra Galea-Outon, Alicia Prieto, Jorge Barriuso

**Affiliations:** Centro de Investigaciones Biológicas Margarita Salas (CIB), Consejo Superior de Investigaciones Científicas (CSIC), Ramiro de Maeztu 9, 28040 Madrid, Spainaliprieto@cib.csic.es (A.P.)

**Keywords:** bioplastic, starch, lignocellulosic biomass, *L. plantarum*, biofilm

## Abstract

Polylactic acid (PLA) is the main biobased plastic manufactured on an industrial scale. This polymer is synthetized by chemical methods, and there is a strong demand for the implementation of clean technologies. This work focuses on the microbial fermentation of agro-industrial waste rich in starch for the production of lactic acid (LA) in a consolidated bioprocess, followed by the enzymatic synthesis of PLA. Lactic acid bacteria (LAB) and the fungus *Rhizopus oryzae* were evaluated as natural LA producers in pure cultures or in fungal–lactobacteria co-cultures formed by an LAB and a fungus selected for its metabolic capacity to degrade starch and to form consortia with LAB. Microbial interaction was analyzed by scanning electron microscopy and biofilm production was quantified. The results show that the fungus *Talaromyces amestolkiae* and *Lactiplantibacillus plantarum* M9MG6-B2 establish a cooperative relationship to exploit the sugars from polysaccharides provided as carbon sources. Addition of the *quorum sensing* molecule dodecanol induced LA metabolism of the consortium and resulted in improved cooperation, producing 99% of the maximum theoretical yield of LA production from glucose and 65% from starch. Finally, l-PLA oligomers (up to 19-LA units) and polymers (greater than 5 kDa) were synthetized by LA polycondensation and enzymatic ring-opening polymerization catalyzed by the non-commercial lipase OPEr, naturally produced by the fungus *Ophiostoma piceae*.

## 1. Introduction

The worldwide production of conventional plastics is increasing every year (368 million tons in 2020) [1], and one-third is unintentionally released into the environment, where it remains for centuries due to its recalcitrance. These polymers can be partially degraded and, eventually, become part of the trophic chain, affecting biodiversity. Bioplastics can be excellent alternatives to conventional materials derived from fossil sources because they can be obtained from plant biomass, a renewable resource that ensures a sustainable production [2]. Among them, biodegradable polylactic acid (PLA) accounts for 18.7% of the global production of bioplastics and is the main biobased plastic manufactured on an industrial scale [1]. The properties of PLA are similar to those of fossil-based plastics, and it can partially replace polypropylene, polyethylene terephthalate and polystyrene.

The platform molecule for PLA synthesis is lactic acid (IUPAC names: 2-hydroxypropionic acid or 2- hydroxypropanoic acid), a chiral molecule that can be found as d- or l- enantiomers. PLA is an aliphatic polyester obtained either by polycondensation of monomeric lactic acid (LA) or by ring-opening polymerization of its cyclic dimer, denominated lactide. However, the industrial demand for LA goes beyond PLA manufacture, since it is used as a flavor-enhancing agent, acidulant, preservative and intermediate for many end products in the food, textile, chemical and pharmaceutical sectors [3]. 

LA can be synthetized by chemical methods or by microbial fermentation [4]. The chemical route is expensive, gives a racemic mixture as reaction product and is dependent on by-products of fossil fuel industries [5]. Microbial fermentation, on the other hand, yields an optically pure l- or d-lactic acid by selecting the appropriate microorganism and growth conditions [6]. Sugar transformation to lactic acid can be catalyzed by many bacteria or fungi through homofermentative, heterofermentative or mixed pathways. The formation of by-products and the high cost of pure sugar substrates are the main drawbacks of the microbial production of lactic acid. The production cost can be diminished, however, by using cheap renewable sources as those derived from lignocellulosic biomass or other waste products [7]. The proportion of l- or d-lactic acid enantiomers affects the properties of PLA: polymers with a higher l-content tend to crystalize and have a higher melting temperature, whereas those with lower optical purity are amorphous and have a lower melting temperature [8,9].

The profitability of the biotechnological production of LA is linked to the cost of the fermentation substrate. The use of inexpensive carbohydrate-rich waste from industrial activities (e.g., starch or lignocellulose residues) represents an elegant solution to this issue, contributing to the development of circular processes [10]. Starch is a reserve polysaccharide in plants, and many starch-based waste feedstocks can be used as substrates for microbial fermentation. This semicrystalline polysaccharide is formed by the combination of two α-1,4-d-glucans, one of them linear (amylose) and the other one a *O*-6-branched (amylopectin) that is coiled in the form of a helix. Due to its highly ramified structure and to the establishment of intermolecular interactions, starch is hard to solubilize in water [11]. Lignocellulosic biomass is more complex than starch and plays an essential structural role in plants due to the association of its individual components: cellulose, hemicellulose and lignin [12]. Cellulose, the most abundant polymer on earth, is a homopolymer made of linear chains of β-(1,4)-d-glucose highly juxtaposed by parallel H bonds, making it extremely resistant and insoluble in water. Hemicelluloses are a family of branched heteropolymers that can contain several monosaccharides (xylose, mannose, arabinose, glucose, galactose, etc.). Lignin is an amorphous heteropolymer of phenylpropane units that is really complex and recalcitrant to degradation [13]. The polysaccharides contained in these polymeric residues must be degraded to simple sugars, to make their use as carbon sources feasible for microbial fermentation. Starch is degraded by α-amylases and α-amylosidases and is an easier substrate for microbial fermentation than lignocellulose, where the concerted action of a pool of β-glycosidases, β-1,4-endo- and exoglycosidases and other enzymes is needed to achieve its complete assimilation. In both cases, filamentous fungi are an outstanding source of deconstructing enzymes [14,15].

Industrial microbial fermentations are typically performed by a single microbial strain in a bioreactor. In these conditions, a pretreatment (physical, chemical and/or enzymatic) of the substrates is usually needed to make the sugars accessible to the microbial strains. However, the utilization in a consolidated bioprocess of combinations of different microorganisms with different metabolic abilities, such as bacteria and lignocellulosic fungi, is rarely explored. The difficulty of this approach lies in the coexistence of two or more microorganisms over time in a culture in which cooperative interactions occur: where the different components of the consortium exchange metabolites. The effect of environmental factors such as growth temperature, pH tolerance, duplication times and metabolic capacities of isolated microorganisms should be considered so that different microorganisms establish productive social relationships.

Microbial behavior is often influenced by the interaction between species forming consortia. When cooperative interactions occur in natural environments, the community takes advantage of the individual capacities of each type of organism and relations among species are usually coordinated through metabolic interactions and signal molecules. One of the main signaling mechanisms is denominated *quorum sensing* (QS), a cell-to-cell communication process described in bacteria and fungi that involves the cell-density-dependent release of specific chemical signals (QSMs) that diffuse and accumulate in the environment [16]. QS has been studied in bacteria, including LAB [17]; however, there are few works on QS in fungi [18]. The first QS signaling molecules described were N-acyl- homoserine lactones that act on many Gram-negative bacteria. Recently, it has been shown that some Gram-positive organisms such as *Lactobacillus plantarum* also respond to these molecules [19]. Furthermore, farnesol (a terpene) and dodecanol (an alcohol) induce different responses among fungi [20] and affect biofilm formation differently in consortia of *Ophiostoma piceae* and *Pseudomonas putida* [21]. Biofilms allow microorganisms to adhere and form a resistant new ecological niche [22]; the regulation of biofilm formation by QS in bacteria and fungi has already been demonstrated [21]. These types of structures, which are common in nature, can be very interesting for the development of consortia for industrial biotransformations.

There are three extended methods to produce PLA polymers. Direct condensation of LA molecules produces low molecular weight prepolymers that can be further converted into PLA through chain coupling agents. In the second procedure, the cyclic dimer of LA, lactide, is condensed into PLA through ring-opening polymerization (ROP). This method decreases polydispersity and leads to a higher quality PLA. Finally, the third method is the direct polycondensation of LA through an azeotropic dehydration of lactic acid [5]. Both direct condensation and ROP can use chemical catalysts or enzymes (lipases and some proteases), which have the main advantage of being environmentally friendly. Lipases are α/β hydrolases whose natural role is the breakdown of ester bonds in aqueous environments. However, in low (or null) water-activity media they catalyze synthesis reactions [23]. Several commercial lipases have already been evaluated in PLA synthesis, both in bulk and in organic solvents such as toluene [24], but the search for new enzymes and green solvents that can facilitate these reactions is a topic of interest. Ring-opening polymerization of lactide is the most used industrial method for PLA production. The catalyst chosen is usually tin(II) octoate (stannous bis(2-ethylhexanoate): Sn(Oct)_2_), whose removal should depend on safety data and food and medical application legislation [25]. Thus, the use of enzymatic catalysts should be a way to overcome this limitation.

This study explores the one-pot fermentation of agro-industrial waste streams to produce LA through a consolidated bioprocess involving an artificial interkingdom consortium between a cellulolytic or an amylolytic fungus and a lactic acid bacteria [7,26]. The eukaryotic partner is expected to degrade the residues supplied as carbon sources, supporting nourishment for the bacteria, which would produce LA. To promote the bacterial–fungal cooperation and its efficiency for LA production, we studied the effect of three QSMs, farnesol, 3-oxo-C12-HSL and dodecanol, to deepen the knowledge of the mechanisms that explain their biological action. Finally, a recombinant fungal lipase was tested to produce PLA by direct condensation of LA or by the enzymatic ring-opening polymerization of lactide.

## 2. Materials and Methods

### 2.1. Strains, Culture Media and Chemicals

Fungi from the Center for Biological Research (CIB) of the Spanish National Research Council (CSIC) collection and lignocellulolytic fungi selected from the bibliography, available in different public collections, were used for this study (Table 1). They were grown in potato dextrose agar plates at 28 °C for 3 days. Two available strains of *L. plantarum* were used: WCFS1 (described as amylolytic) and M9MG6-B2 (non-amylolytic). They were grown at 28 °C on De Man, Rogosa, and Sharpe (MRS) broth for 1 day [27]. 

l-lactic acid, l-lactide and the QSMs used, *trans,trans*-farnesol, 3-(oxododecanol)-homoserine lactone and 1-dodecanol, were provided by Sigma-Aldrich (St. Louis, MO, USA). Absolute ethanol was used to resuspend *trans,trans*-farnesol and 1-dodecanol. Homoserine was prepared in 0.01% glacial acetic in acetonitrile. Ethyl lactate was provided by Corbion (Purasolv).

### 2.2. Fungal Screening

The fungi listed in Table 1 were tested for their metabolic capacities to degrade starch. Fully colonized agar discs of 8 mm diameter were placed in the center of two Czapek Dox agar plates [28], with 1.5% (*w*/*v*) starch or without a carbon source. The plates were incubated at 28 °C for 5 days in duplicate. After that, the growth diameter of the mycelium was measured, and the clearing halo diameter was analyzed by staining with 5% (*v*/*v*) diluted Lugol’s iodine (Merck) to evaluate the starch degradation [29].

### 2.3. Growth Conditions

Firstly, bacterium–fungus combinations were tested in liquid medium: GPP (Glucose-Proline-Phosphate) minimal medium [21], Czapek’s Dox [29], CDM (chemical defined medium) [30] and plantarum minimal medium (PMM5) [27], all with glucose as a carbon source. In subsequent experiments, we used MRS broth (Condalab) with glucose as a control medium and MRS broth without dextrose and beef extract (Condalab) with 1% (*w*/*v*) starch as the carbon source. In all cases, 15 mL tubes were filled with one-third of its total volume.

One representative fungus (*T. amestolkiae*), selected for its amylolytic capacity in the previous screening, was chosen to determine the optimal growth conditions in co-cultures. A 10:1 fungus–bacteria inoculum ratio was used according to previous results [21]. A preinoculum of the selected fungal strains was grown in GPP medium at 28 °C and 200 rpm for 4 days and filtered through Miracloth to separate spores from hyphae. The spores were counted in a Thoma counting chamber under a light microscope (Zeiss, 40× magnification), and a spore solution was prepared to inoculate at a final concentration of 5 × 10^6^ cells/mL. The bacterial inoculum was 10-fold lower (5 × 10^5^ cells/mL). Both bacterial and fungal microorganisms were co-inoculated in MRS broth with 2% glucose or 1% (*w*/*v*) pure starch from Sigma Aldrich as a carbon source and incubated at 0, 20 or 200 rpm at 28 °C. Experiments were performed in duplicate. Microbial growth was observed under the light microscope (Zeiss, 40× magnification) after 24 h and 72 h of incubation. LA production was measured using the “l-Lactate and d-Lactate Assay Kits” (Sigma-Aldrich), as described below.

The experimental design to test different QSMs and microbial combinations was conducted at 200 rpm and 28 °C for 72 h, filling one-third of the volume of the 15 mL tubes, and, when necessary, a QSM (farnesol, dodecanol or 3-oxo-C12-HSL) was added at a final concentration of 100 µM, based on previous works [18]. MRS broth with glucose was used for control experiments, whereas when starch or industrial residues (1% *w*/*v*) were tested, MRS broth without dextrose or beef extract was used. LA production and starch consumption were analyzed by HPLC, as described in Section 2.5. Experiments were performed in triplicate.

### 2.4. Agro-Industrial Residues and Pre-Treatment

After setting the conditions for transformation of pure commercial starch in LA, two industrial wastes were finally assayed as C sources to obtain LA. The starch residues were provided by Espafrima S. L. (Madrid, Spain) and beer bagasse was supplied by La Cibeles S.A. (Madrid, Spain). Pretreatment of beer bagasse was performed as follows: it was ground and sieved (0.42 mm), then treated with 5% (*w*/*v*) NaOH for 1 h at 50 °C in agitation; next, it was cooled and neutralized up to pH 6-5, centrifuged to remove most of the liquid and then lyophilized.

### 2.5. LA Quantification, Starch Degradation and HPLC Analysis

LA production was preliminarily analyzed using the colorimetric assay, “l-Lactate and d-Lactate Assay Kits” (Sigma-Aldrich), an LA standard curve was performed in a SpectraMax Plus 384 spectrophotometer. LA production was also analyzed by HPLC using an Agilent 1200 series equipped with a diode array detector set at 210 nm connected in series with a refractive index detector (RI 830). The samples were filtered and 10 µL was injected and analyzed isocratically at 0.5 mL/min with 5 mM H_2_SO_4_ as the mobile phase in a Bio-Rad Aminex HPX 87H column (300 × 7.8 mm i.d.) at 55 °C. To determine starch consumption, the G Protocol (for samples containing resistant starch in suspended form) of the Total Starch Assay Kit (AA/AMG) (Megazyme) was followed. The released glucose was further analyzed by HPLC as described above for LA detection, and the starch concentration was calculated by applying a correction factor of 0.9 (Mw polymerized glucose/Mw free glucose). LA and glucose standard curves were injected in the same conditions to interpolate peak areas and quantify concentrations. Yields of LA were calculated by assigning as 100% LA the maximum theoretical production of this compound from 2% glucose (about 220 mM) and 1% starch (110 mM).

### 2.6. Biofilm Quantification

To measure the amount of biofilm formed, 500 µL of the individual or mixed cultures in MRS medium with starch as the C source were placed in a sterile 24-well plate and incubated at 28 °C and 200 rpm (Minitron AG CH-4103 Bottmingen, Infors HT). A final concentration of 5 × 10^6^ cells/mL of the fungal spore solution was added and incubated for 24 h. Then, 100 µM dodecanol and the LAB inoculum (5 × 10^5^ cells/mL) were added and incubated for 24 h. Planktonic cells and hyphae were then collected to determine cell density by turbidimetry (DO_600nm_, UV-1900i UV-Vis Spectrophotometer Shimadzu). After discarding the supernatants, the plate was washed with distilled water and 0.1% (*w*/*v*) crystal violet was added and incubated with the biofilm for 15 min. After washing with distilled water, the plates were dried at room temperature and 1 mL of ethanol 96% was added for destaining the biofilm. The ethanolic solution was recovered and diluted as required to measure the absorbance at 590 nm. Biofilm cell density was deduced from these values compared with A_590nm_ of clean 96% ethanol.

### 2.7. Sample Preparation for Scanning Electron Microscopy (SEM)

The most efficient consortium producing LA in the commercial starch medium was selected for SEM analysis. Sterile PTFE 0.22 µm filters (Fluoropore, EMD Millipore) were placed in the 24-well plate cultures to let biofilms attach to them for 48 h. Then, the filters were fixed in a 2.5% (**v*/*v**) glutaraldehyde solution overnight at 4 °C, washed in distilled water twice for 5 min and successively desiccated in 30%, 50%, 70%, 80%, 90% and 100% ethanol (10 min each). Finally, the filters were subjected to a critical point and metallization processes at the Spanish National Centre for Electron Microscopy (Universidad Complutense de Madrid) to take the final images in a JEOL 6400 SEM.

### 2.8. PLA Production by Enzymatic Polycondensation of LA

A recombinant form of the versatile lipase/esterase from *O. piceae,* produced in *Pichia pastoris* (OPEr), was selected to produce PLA by direct condensation of the monomers of l-LA [31]. The experiments were performed in duplicate. Polycondensation reactions were carried out in hermetic vials containing 20 µL milli-Q H_2_O, toluene up to 2 mL, 1.25 mmol of l-lactic acid (l-LA) and 10 U of enzyme (activity against *p*-nitrophenyl butyrate) [31]. The vials were incubated at 40 °C in an orbital shaker at 1000 rpm (AccuTherm—Labnet International, Inc. Global, Woodbridge, NJ, USA) for 6 days and then lyophilized. To dissolve the PLA formed, 0.5 mL chloroform was added and the samples incubated for 10 min at 60 °C. After adding 0.5 mL milli-Q and incubating for 10 min at room temperature, the organic and aqueous phases were separated. The synthesis of PLA (organic phase) was assessed by MALDI-TOF and the unreacted LA (aqueous phase) by GC/MS at the Gas Chromatography Service of CIB-CSIC.

### 2.9. PLA Production by Enzymatic Ring-Opening Polymerization of Lactide

OPEr was also tested as a catalyst for eROP reactions of l-lactide. For this reaction, the enzyme was lyophilized and maintained together with the substrate and solvent in a desiccator with P_2_O_5_ until used. Reactions were conducted in bulk or in 0.5 mL ethyl lactate as solvent in sealed vials in the following conditions: 15 U of OPEr, 100 °C, 800 rpm and 150 mg of l-lactide. Considering the insolubility of l-lactide, we assayed three different procedures to add it to the reaction mixture: 3 pulses of 50 mg at 0 h, 24 h and 48 h (A), an initial pulse of 100 mg followed by a second pulse of 50 mg after 48 h (B) or all the substrate in a single pulse at the beginning of the reaction (C). Samples were taken every 24 h for 3 days, prior to the addition of lactide when that was the case. Substrate conversion was determined by ^1^H-NMR, calculating the lactide:PLA ratio from the areas corresponding to the following chemical shifts: 5.0 ppm for lactide and 5.2 ppm for PLA. All NMR samples were directly prepared in 5 mm NMR tubes in chloroform-d containing 0.1% TMS as a reference at 298 K. The standard Bruker pulse program, zg for ^1^H-NMR, was performed on a Bruker AVIII spectrometer equipped with a probe TXI 600MHz S3 5 mm with Z-gradient. The data were processed with Topspin 3,5 pl5 software.

## 3. Results and Discussion

### 3.1. Selection of Amylolytic Microorganisms

As a first step, eleven fungi (Table 1) were tested for their metabolic capacities to degrade commercial starch. These species were selected from public collections for having genes that encode amylases or for their known capacity to degrade polysaccharides from plant biomass. To determine the presence and extent of amylolytic activity and their growth in starch as the only C source, the strains were cultivated at 28 °C for 5 days in Czapek Dox agar medium (pH 7) with 1.5% (*w*/*v*) pure starch from Sigma, measuring the lightening halo and the diameter of the colonies. The halo around the colony corresponds to starch degradation in this area by extracellular enzymes produced by the microorganism. Cultures of the microorganisms grown in identical conditions, but without a C source, were used as negative controls.

Some of the fungi evaluated grew in the control plates, suggesting that they may produce extracellular agarase activity. However, the observation of a higher growth in the starch plates and the appearance of degradation halos indicates that there was extracellular amylase activity secreted. According to the data in Table 1, the halos were not observed in some of the species evaluated or were very small in the conditions assayed. The *A. oryzae* strains tested were good producers of α-amylase activity [32], whereas for *Penicillium* strains, α-amylase was detected in *P. citrinum* [33] but not in *P. frequentans* [34]. The three species that best combined good growth on starch plates with the largest inhibition halos were *R. oryzae,* a well-known producer of amylases [35], followed by *T. amestolkiae* and *O. piceae.* The two last species are known for their activity on plant polysaccharides [36] and lipids [37], respectively, and their amylolytic activity is reported here for the first time, evidencing the great potential of these microorganisms for biomass modification. In view of these results, and although it must be taken into account that the standard conditions used for this screening may not be optimal for some of the species, these three fungi were selected for subsequent experiments.

On the other hand, the two LAB selected were the homofermentative *L. plantarum* strains M9MG6-B2 and WCFS1. This last species was chosen because its genome contains a gene that codes for amylase, a very rare trait among LAB. However, when the amylolytic potential of this strain was assessed, no amylase activity was detected. Despite this result, we decided to keep it in further experiments to verify its behavior in consortia compared with the non-amylolytic strain.

### 3.2. Optimization of Growth Conditions for Interkingdom Consortia

After selecting the fungal strains, the growth of the microorganisms involved in the study was tested in different minimal media with glucose as the carbon source (GPP, Czapek Dox, CDM and PMM5). The three selected fungi grew up in all of them, but LAB growth was very slow (not shown). Since LAB are frequently cultivated in MRS complete medium, the culture of the selected fungi in this medium was assayed, with good results. Therefore, MRS was selected for further experiments. In these trials, LA concentration was measured using the “l-Lactate and d-Lactate Assay Kits” (Sigma-Aldrich) in order to confirm that the selected growth conditions do not interfere with its production.

The effect of agitation in the co-cultures was analyzed using only *T. amestolkiae* as a representative of the three amylolytic fungi. Optimization of this parameter is especially relevant due to the different requirements of fungi and LAB, since the selected fungi need higher aeration, whereas the bacterial fermentations occur in low-oxygen cultures. Fortunately, lactic fermentation is one of the few fermentative processes that can take place in the presence of oxygen, although it is known that LA yield decreases in favor of biomass growth [38]. Different agitation conditions (static, 20 rpm and 200 rpm) were tested with both LAB strains in pure cultures and in co-cultures with *T. amestolkiae*. LA production was measured after 24 h incubation, and the co-culture was observed by light microscopy after 72 h (Figure 1).

The production of l-lactate in all conditions (d-lactate was not detected) demonstrated that lactic fermentation occurred in each case, usually being higher in static cultures (not shown). The non-amylolytic strain M9MG6-B2 was a better l-LA producer than WCFS1, and the production observed in agitated pure bacterial cultures depended on the speed. These results were similar in co-cultures with the fungus, which indicated that the presence of *T. amestolkiae* does not exert a negative influence on LA production by the bacterial partner.

Regarding fungal growth, in static cultures only conidiospores were observed (Figure 1A), whereas there was development of hyphae in shaken cultures that was more abundant at 200 rpm (Figure 1C). Therefore, we selected 200 rpm as the best agitation for consortia to favor fungal growth and still produce high levels of LA, since fungal spores are metabolically inactive [39]. 

### 3.3. LA Production from Glucose and Starch and the Effect of QSMs

Once the agitation (200 rpm) and the basal medium (MRS) were selected from the previous experiment, using *T. amestolkiae* as a representative fungus we evaluated the efficiency of the synthesis of LA in fungus–bacteria consortia formed by one of the three preselected fungi in combination with the two bacterial strains. As control conditions, the production of lactic acid in pure cultures of the bacterial strains and *R. oryzae* (this fungus was also capable of releasing lactic acid naturally) was also analyzed. To test the effect of adding *quorum sensing* molecules, cultures with or without farnesol, dodecanol or 3-oxo-C12-HSL were compared. LA production yield was analyzed by HPLC as described in Section 2.5 and calculated with respect to the theoretical maximum releasable from the substrate (100%) in media with glucose (Figure 2) or pure commercial starch (Figure 3) as the carbon source.

As shown in Figure 2, the production of LA in a control medium with glucose and without added QSMs varied from 15% in *R. oryzae* (labeled as R in Figure 2) to 98% in *L. plantarum* WCFS1 alone (labeled as W) and in its consortium with *T. amestolkiae* (TW); despite most of these differences being not statistically significant, trends could be observed. The effect of supplementing with one of the QSMs was different in each case, although in most cases it was neutral or positive for LA release. The most relevant enhancements were due to farnesol (in W, TW and TM). A clear decrease in LA was observed in the consortium *R. oryzae*–*L. plantarum* M9MG6-B2 (RM) with farnesol, the consortium *R. oryzae*–*L. plantarum* WCFS1 with dodecanol and *L. plantarum* WCFS1 alone with dodecanol. *R. oryzae* was unsensitive to any of the signal molecules.

The low production of lactate by *R. oryzae* alone is remarkable [40]; however, it is interesting to note that up to 7 mM succinic acid was also detected in the culture medium, which means a metabolic change to secondary products of the TCA cycle. The production of this acid in trace amounts has been previously reported [41,42], and the higher concentrations detected in the current work might be due to transformation of components of the MRS medium into this by-product. Its presence may indicate a deviation of the metabolic fluxes affecting the synthesis of LA, and its production is not affected by any of the QSMs used, the combinations with bacteria or the C source.

As expected, the maximum LA production from starch (65% corresponding to 72 mM, Figure 3) was, in general, much lower than that from glucose (99% corresponding to 200 mM, Figure 2) since the theoretical maximum yield from glucose is twice that from starch. In terms of percentage regarding the initial substrate concentration, production from glucose is much more efficient since starch consumption requires high amylase activity to depolymerize it to monosaccharides assimilable by the LA producer. In this sense, the disappearance of starch was directly correlated to LA concentration. In control cultures without QSMs, *R. oryzae* (R) produced a moderate amount of LA (around 30% of the maximum theoretical value), confirming the activity of its extracellular amylase and its intracellular l-lactate dehydrogenase. However, both LAB cultures produced low concentrations of LA. Although the value was slightly higher in *L. plantarum* WCFS1, the poor amylolytic capacity of this strain in these conditions was corroborated. Thus, these bacteria should cooperate with one of the fungi able to metabolize starch for a fruitful bioprocess. This was observed in co-cultures of *L. plantarum* M9MG6-B2 with *T. amestolkiae* (TM) and of *L. plantarum* WCFS1 with *R. oryzae* (RW). In the latter case, the LA produced was higher than in pure cultures of the bacteria but smaller than in those of *Rhizopus*, which suggests that both microorganisms compete for the glucose released from starch by fungal amylases, causing a detrimental impact on LA production. 

The addition of QSMs resulted in different effects, as previously observed in cultures with glucose (Figure 2). Supplementation with dodecanol proved to be positive in six of the nine cultures, although the most striking effect on LA levels was noticed in the TM consortium (64%), tripling the value of its control without QSM and doubling those observed with farnesol and for *R. oryzae* (R) with dodecanol or farnesol. It is also interesting to note that combinations of *R. oryzae* and *T. amestolkiae* with *L. plantarum* M9MG6-B2, which lacks amylase activity, generated higher production than with the amylolytic strain WCFS1. On the other hand, despite the fact that the presence of dodecanol has been described in the bioactive fraction of an isolate of *R. oryzae* [43], no significant effects of this or other QSMs were observed in cultures of this strain, confirming the data obtained in the medium with glucose.

In the case of the combinations of *O. piceae* with any of the LAB in glucose (Figure 2) and in some starch cultures (Figure 3), the LA yield seems to decrease compared to their controls. Thus, we can consider that these QSMs have an effect on *O. piceae* that indirectly impacts LA production; however, we cannot affirm that quorum quenching occurs, as it is unclear whether the fitness of the community is affected. In fact, previous studies showed pro-biofilm effects of these three QSMs in *O. piceae* [21].

The effect of QSMs could influence the degradation of starch or the fermentation of glucose to LA by LAB, but comparison of the data obtained in glucose and starch did not reveal big differences to shed light on this matter. In the most suitable consortium among those studied in this work there is a clear cooperative relationship between partners, with *T. amestolkiae* decomposing starch and *L. plantarum* M9MG6-B2 converting glucose into LA and, probably, releasing other substances that the fungus uses to grow. This cooperation is improved by the inductor effect in the microbial metabolism of farnesol in the case of glucose medium and dodecanol in the case of the medium with starch. 

According to our results, the addition of 3-oxo-C12-HSL had little impact on LA production. Previous studies with this molecule, typically associated with QS interactions in Gram-negative bacteria [44], demonstrated that the Gram-positive *L. plantarum* WCFS1 is also responsive to it and reported the over-expression of several oxidative enzymes [19]. We have not observed this behavior, and our data also indicate that this QSM does not affect LA production by *L. plantarum* M9MG6-B2. There are not studies about the response of this strain to acyl-homoserine lactones. 

Regarding farnesol, it is known that this QSM affects yeast–hyphae transition, biofilm formation and enzyme secretion in some fungi, such as *Candida albicans* [45], *O. piceae* [18] and *T. amestolkiae*. However, the outcomes of this work show that farnesol exerted a negative effect in the fermentations with glucose (Figure 2) and in starch co-cultures (Figure 3) of *R. oryzae* with *L. plantarum* M9MG6-B2 (RM) (although this was not observed when the microorganisms were independently grown). Further studies are needed to unveil how farnesol acts on the associations and molecular mechanisms of these microorganisms.

Dodecanol is considered a structural analog of dodecanoic acid, a bacterial QSM, and has structural similarity to farnesol and 3-oxo-C12-HSL (C12 motifs/backbones). These molecules may interact with similar receptors, and its role in fungal morphogenesis has been reported [46]. In addition, most of the cultures in starch (Figure 3) with dodecanol had increased LA production compared with controls. This can be interpreted as a response that enhances the degradation of the C source by increased extracellular enzymatic activities and/or a more efficient fungus–bacterial association. In our study, this QSM exerts the best effect on LA production from starch. The co-culture of *T. amestolkiae* and *L. plantarum* M9MG6-B2 with 100 µM dodecanol yielded 65% from soluble starch (LA production (g)/starch consumed (g)) (Figure 3). This strain combination was selected for further experiments.

### 3.4. Biofilm Formation

After selecting the best performing consortium (*T. amestolkiae*–*L. plantarum* M9MG6-B2) and signal molecule (dodecanol) in starch, we evaluated the ability of these microorganisms to form biofilms. This capacity can be influenced by the establishment of social interactions between microorganisms and influences the colonization of the biomass to be degraded. 

The experiment was carried out with the consortium and in individual cultures (controls) with and without dodecanol, measuring the amount of crystal violet (A_590 nm_) fixed by the cells adhered to the bottom of the wells (Figure 4).

The amount of biofilm in the *Talaromyces* culture was higher than in the LAB culture, and the addition of dodecanol increased biofilm formation in the consortium. Some studies have shown the ability of *L. plantarum* species to form biofilms [47], something quite common in LAB since many of them are adhered to their natural habitat, the gastrointestinal wall. However, this is the first report on biofilm formation by *T. amestolkiae* alone or in consortium with a *L. plantarum* strain.

The structure of biofilms was examined by scanning electron microscopy, which allowed observing the adhesion of bacterial cells to fungal hyphae in the presence of dodecanol (Figure 5). This suggests that this QSM may increase the association of both microorganisms.

### 3.5. LA Production from Industrial Residues Catalyzed by a Fungus–Bacteria Consortium

As we have seen above, the consortium *T. amestolkiae*–*L. plantarum* M9MG6-B2 with dodecanol was the best option for an effective production of LA from pure commercial starch. Moreover, the association of both strains was confirmed by SEM (Figure 5). Thus, this combination was further tested with two real agro-industrial residues (starch from a potatoes chip factory and bagasse from a beer factory) in separate cultures without any other carbon source. The production of LA from these two sources is presented in Figure 6.

In the case of the potato residue, there is a direct relationship between starch consumption and LA production, which reaches a maximum at around 24 h of incubation (Figure 6A). Then, starch levels do not continue going down and LA is consumed. Surprisingly, LA production from bagasse at the same incubation time was higher and seems to continue to increase, albeit more slowly. Bagasse is supposed to be more difficult to decompose than starch, but *T. amestolkiae* has been described as an excellent producer of cellulases and hemicellulases, which could explain the good conversion of bagasse into fermentable sugars [48]. In addition, the complex composition of this substrate could provide components that favor the growth of the consortium and explain the higher yields.

### 3.6. Enzymatic Synthesis of PLA

The recovery of LA obtained by microbial fermentation is out of the scope of this work. However, the enzymatic synthesis of PLA bioplastic is an interesting objective, since current manufacturing procedures use toxic solvents and/or high temperatures. In this context, a recombinant form of the sterol esterase/lipase naturally secreted by *O. piceae* and produced in *P. pastoris* (denominated OPEr) has been extensively studied for its efficiency and versatility [37]. Since this enzyme has shown better properties than several commercial lipases in esterification, transesterification and hydrolysis [49], we have evaluated its efficiency for PLA synthesis, either from lactic acid (by direct polycondensation) or from its cyclic dimer, lactide (by enzymatic ring opening polymerization, eROP). The first reaction involves the release of water as a product of the condensation of LA monomers, which deviates the reaction to hydrolysis once a certain amount of PLA has been formed. As a result, the PLA produced by this procedure usually has a low polymerization degree [50].

The samples were qualitatively analyzed by MALDI-TOF (Figure 7), which allowed detecting patterns characteristic of oligomers of different lengths resulting from addition of 72 Da, which corresponds to the mass of LA residues. The signals observed in the spectrum of the negative control (without enzyme) were attributed to the Na^+^ adducts of chains from four (329 Da) to seven LA units (547 Da), as LA naturally forms small chains. However, the spectra of samples treated with OPEr showed bigger LA chains, up to 19-LA units (1386 Da), indicating that the condensation reaction occurred.

On the other hand, enzymatic ROP (eROP) is known to produce PLAs of a higher polymerization degree than direct polycondensation. In this case, the substrate for the reaction is lactide, which is a relatively insoluble compound that requires toxic solvents and/or very high temperatures to be polymerized in bulk [51]. Therefore, apart from testing reactions in bulk, we have also analyzed the suitability of using ethyl lactate as a green solvent for the reaction catalyzed by OPEr. PLA synthesis was followed on this occasion by ^1^H-NMR (Figure 8). As no peaks from short chain oligomers were observed by MALDI-TOF in a m/z range from 200 Da to 5 kDa, this suggested that longer polymers had been synthesized.

The lactide:PLA ratios were similar regardless of how the substrate was incorporated to the reaction mixture (Figure 8A–C). Thus, fractioned addition of lactide does not translate to an improvement of the reaction’s efficiency. This is different to what is observed in other synthetic reactions catalyzed by lipases [52]. From the same plots, it can be deduced that reactions take longer to begin in ethyl lactate than in bulk; however, the final conversion at day 3 is virtually complete in ethyl lactate, whereas there is lactide left in bulk reactions. Thus, the solubility of this substrate in ethyl lactate will probably favor its complete polymerization. 

In terms of substrate conversion only, these results are good compared with some previously published works. Hans et al. [52] catalyzed the eROP of d-lactide with Novozyme 435 (N435) (12.5%, wt) in toluene at 70 °C for 3 days, and ^1^H-NMR analysis revealed a substrate conversion of 33%, quite lower than the conversion obtained here. Chanfreau et al. [53] attempted the eROP of l-lactide in a green solvent as well, the ionic liquid 1-hexyl-3-methylimidazolium hexafluorophosphate [HMIM][PF6] mediated by N435 (10%, wt). The highest l-PLA yield (63%) was attained after 7 days at 90 °C. Nevertheless, Duchiron et al. [54] found that N435 (10%, wt) converts 98% of d-lactide at 70 °C for only 2 days in toluene, as does the enzyme LBC with d-lactide at 90 °C. They also found that using triethylamine, a polar aprotic amino base, as a co-solvent leads to five to six times faster reactions. Therefore, studying eROP reactions in the presence of this kind of activators could be a promising way to achieve cost-effective processes. 

From these results, it can be concluded that, among the conditions assayed, the best conversion was obtained adding lactide in a single pulse and with ethyl lactate as solvent. The latter has the added advantage that reactions could probably be conducted at a lower temperature, which is not possible in bulk reactions since the melting point of l-lactide is 92–94 °C.

Our method for eROP presents two economic advantages over other established procedures. First, the recombinant enzyme OPEr is produced in *Pichia pastoris*, described as a very efficient producer of extracellular proteins [55]. In addition, the recombinant enzyme has increased solubility and stability than that secreted naturally by the producing fungus [56]. Also, the l-lactide is soluble at room temperature in ethyl lactate, so, once proven that the molecular mass of the PLA produced is good, eROP reactions could be carried out in much more sustainable conditions. Moreover, with the objective of reducing enzyme production costs, the immobilization of OPEr as *m*CLEAs (magnetic cross-linked enzyme aggregates) is currently being studied in our laboratory, with promising preliminary results. This technique allows the recovery of the insoluble enzymes from the reaction medium using a magnet. This makes the downstream process much easier and cheaper and, if the enzymes prove to be stable in the reaction conditions, they can be reused in consecutive reaction cycles, substantially reducing the industrial cost of the bioprocess. 

## 4. Conclusions

This work demonstrates that *T. amestolkiae* and *L. plantarum* M9MG6-B2 can grow together, establishing a cooperative relationship to exploit the sugars from the polysaccharides provided as C sources. This interaction is favored by the addition of the QSM dodecanol to the culture medium, which induces the LA metabolism of the consortium, produces a closer association of both partners and results in an improved cooperation. In this consortium, the tasks of both microorganisms are divided, with *T. amestolkiae* depolymerizing sugars from the substrate and *L. plantarum* M9MG6-B2 converting glucose into LA. As a proof of concept, lactic acid was also produced from industrial wastes (starch and beer bagasse), although further optimization is needed to improve yields and profitability by exploring further fermentations in different bioreactor setups. Thus, we report here for the first time a consolidated bioprocess that combines a fungus–bacterium consortium and QS induction to transform agro-industrial residues into optically pure l-lactic acid (precursor of PLA). In addition, we have also shown that the fungal versatile lipase OPEr can catalyze the condensation of LA units to give l-PLA oligomers, as well as synthesize longer polymers by eROP of lactide either in bulk or using a green solvent. Even though these experiments should be optimized, the use of microbial and enzymatic biocatalytic tools for bridging the gap from waste to renewable bioplastics is a promising approach to produce essential goods in a green chemistry framework.

## Figures and Tables

**Figure 1 jof-09-00342-f001:**
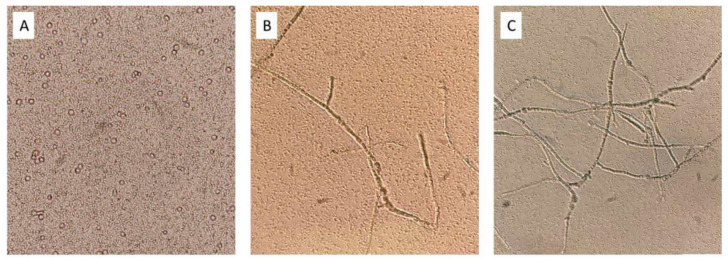
Light microscopy pictures (×40) of *T. amestolkiae*–*L. plantarum* M9MG6-B2 co-cultures after 72 h incubation. (**A**) Static conditions; (**B**) 20 rpm; (**C**) 200 rpm.

**Figure 2 jof-09-00342-f002:**
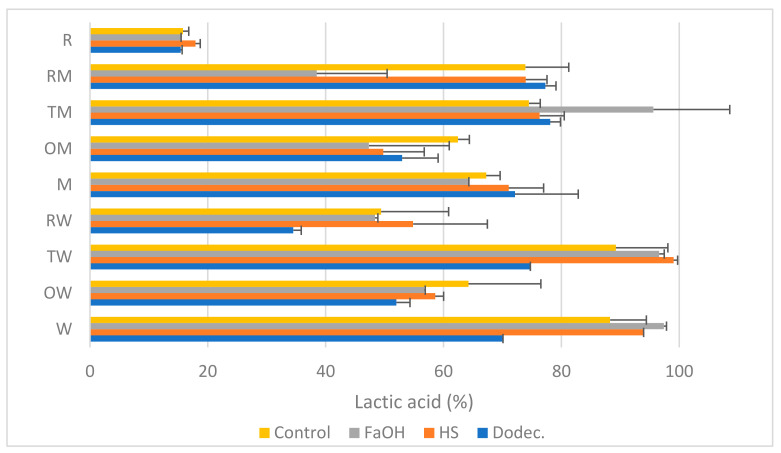
LA production (% of the theoretical maximum) in pure cultures or co-cultures of the microorganisms selected with glucose as carbon source (24 h, 200 rpm, 28 °C). Microorganisms: *R. oryzae* (R), *T. amestolkiae* (T), *O. piceae* (O), *L. plantarum* M9MG6-B2 (M), *L. plantarum* WCFS1 (W) and their combinations. Farnesol (FaOH), 3-oxo-C12-HSL (HS) or dodecanol (Dodec.) were added to the cultures and the LA produced was compared with that from untreated cultures (control).

**Figure 3 jof-09-00342-f003:**
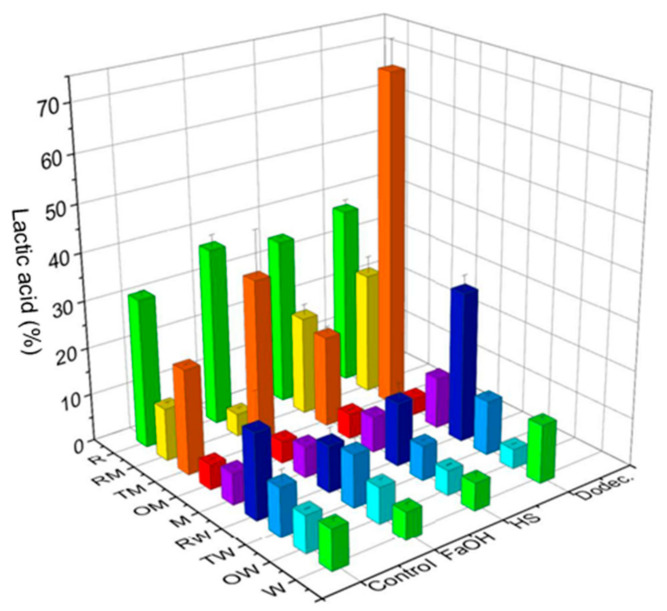
LA production (% of the theoretical maximum) in pure cultures or co-cultures of the microorganisms selected with pure commercial starch as carbon source (24 h, 200 rpm, 28 °C). Microorganisms: *R. oryzae* (R), *T. amestolkiae* (T), *O. piceae* (O), *L. plantarum* M9MG6-B2 (M), *L. plantarum* WCFS1 (W) and their combinations. Farnesol (FaOH), 3-oxo-C12-HSL (HS) or dodecanol (Dodec.) were added to the cultures and the LA produced was compared with that from untreated cultures (control).

**Figure 4 jof-09-00342-f004:**
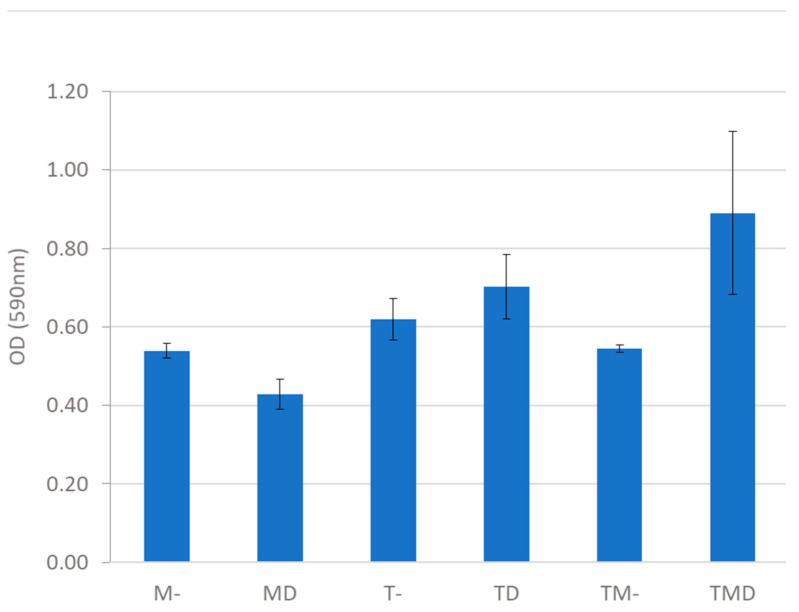
Quantification of biofilm density in MRS medium with starch, induced or not with dodecanol, incubated in a 24-well plate at 28 °C and 200 rpm. *T. amestolkiae* (T), *L. plantarum* M9MG6-B2 (M), 100 mM dodecanol (D), without dodecanol (-).

**Figure 5 jof-09-00342-f005:**
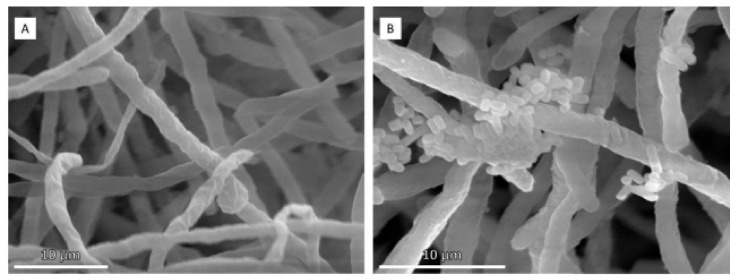
Scanning electron micrographs of *T. amestolkiae*–*L. plantarum* M9MG6-B2 consortia in the absence (**A**) or presence (**B**) of dodecanol.

**Figure 6 jof-09-00342-f006:**
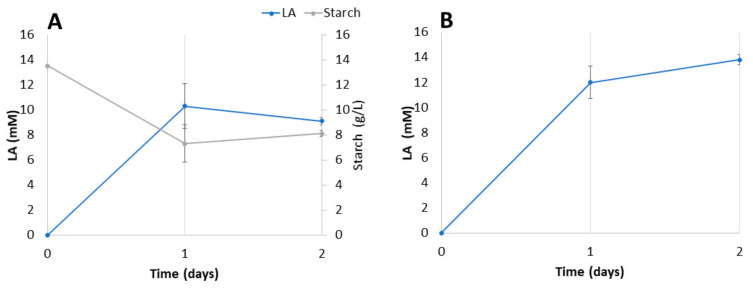
LA production in *T. amestolkiae*–*L. plantarum* M9BG6-B2 consortia growing with industrial starch waste (**A**) or bagasse (**B**) as carbon sources.

**Figure 7 jof-09-00342-f007:**
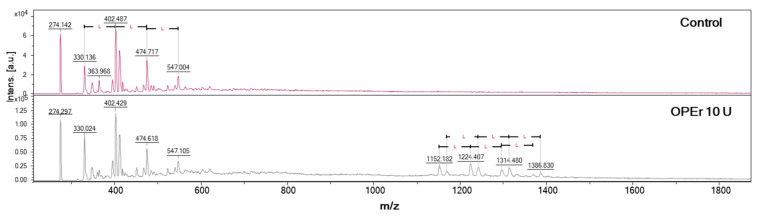
Synthesis of l-PLA by polycondensation of lactic acid catalyzed by OPEr. MALDI-TOF spectra of the chloroform-soluble material extracted from the reaction mixture. The upper plot corresponds to a control reaction without enzyme, and the lower to a reaction with 10 U of enzyme. L represents the incorporation into the chain of one LA unit (72 Da).

**Figure 8 jof-09-00342-f008:**
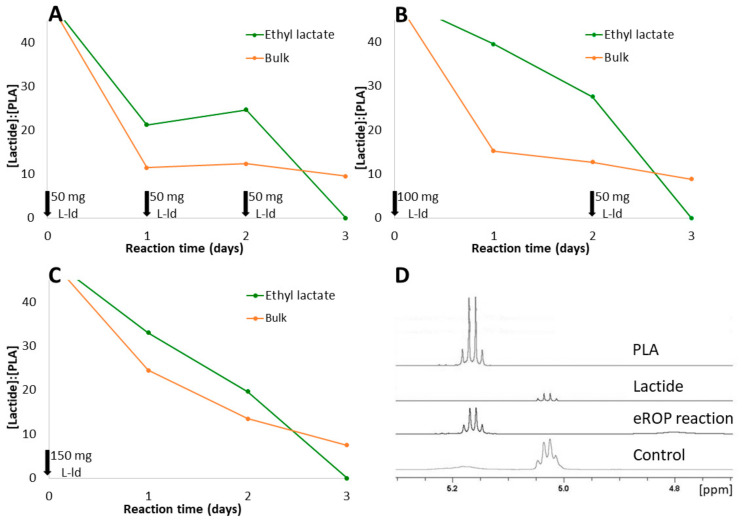
Synthesis of l-PLA via eROP catalyzed by OPEr at 100 °C and 800 rpm. Aliquots of eROP reactions were taken every 24 h and analyzed by ^1^H-NMR. The signals at 5.0 ppm from lactide and at 5.2 ppm from PLA were integrated to calculate the ratios of their areas. Black arrows indicate addition of l-lactide (L-ld) in 3 pulses of 50 mg of lactide every 24 h (**A**); in 2 pulses, one of 100 mg at the start of the reaction and a second one of 50 mg at 48 h of reaction (**B**); or in a single pulse of 150 mg at the beginning of the reaction (**C**). Region between 5.3–4.7 ppm of the ^1^H-NMR spectra of commercial PLA, lactide, an eROP reaction (procedure B in ethyl lactate, t = 3 days) and a control without enzyme (**D**).

**Table 1 jof-09-00342-t001:** Screening of starch degradation by selected fungal strains.

Strain	Lightening Halo	Colony Diameter (cm)	
	(cm)	Control	Starch 1.5%
*Rhizopus oryzae* CBS 111718	Full plate	3	3.5
*Talaromyces amestolkiae* IJFM A795	1	2.3	4
*Ophiostoma piceae* CECT 20416	1	0	4
*Aspergillus oryzae* CECT 20249	1	2	3
*Aspergillus oryzae* ATCC 96995	0.6	0	1.2
*Penicillium citrinum* CECT 20822	0.5	1.5	1.5
*Hypocrea jecorina* CECT 20102	0.3	5.5	2.2
*Aspergillus nidulans* MAD2	0	0	4
*Chaetomium globosum* CECT 2701	0	2.5	2.5
*Penicillium frequentans* IJFM A569	0	2.5	1.5
*Yarrowia lipolytica* W29 CBS 7504	0.8	1.3	1.2

## Data Availability

No data beyond the data presented in the manuscript.

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
