# Peer review of "Fungal–Lactobacteria Consortia and Enzymatic Catalysis for Polylactic Acid Production"

_jof, 2023, doi:10.3390/jof9030342_

Round 1
Reviewer 1 Report
Please note detailed comments on the attached file (I have given recommendations on your manuscript).

Author Response
We are really thankful for to referee 1 for reviewing this manuscript and their helpful comments.
1) If PLA s the main biobased plastic, then why is it only 18.7% of global bioplastics?
In this case we are referring to the total share of bioplastics, either biodegradable (e.g. bio PE accounts for the 9.5 % of the total production) or biobased (e.g. PBAT accounts for the 7.2 %, PBS for the 4.6 % and PHA for the 1.4 % of the total production).
2) "PLA properties...." this is only partly correct. There are limitations compared to plastics. Needs correction
We have corrected the sentence in the abstract as suggested
3) The English needs attention
English has been carefully revised throughout the manuscript making all the changes suggested by the reviewer. For instance, in Lines 55-56 “deeply” and “other” has been eliminated, and so on.
4) Please avoid too much details that are of little relevance to the main message of the paper. The introduction can be reduced by one third without affecting the integrity of the work.
We have reworked the introduction, as the other referees have also suggested. We have avoided too much detail, but we have added information about the chemical synthesis of PLA.
5) Specify the reason for the selection of 100 microM for the QSM
This concentration was selected based on previous works from the group, as is now stated in the text. This concentration mimics a physiological concentration of a signal molecule and avoids possible toxicity of the molecule.
6) Perhaps a rearrangement of the sentences will improve the reading
We have corrected section 2.7 (now 2.6) “Biofilms quantification” as suggested by the reviewer.
7) This is biofilm formed on the surface of 96 well plate. What is the relevance to ...
The biofilm is formed by the attachment of the microorganism to the plastic of the well. We wanted to assess the colonization potential of the microorganisms in a model surface (such as plastic) to evaluate the potential to colonize particles of the agro-industrial wastes.
8) What was the pH?
pH of the plates has been added in the text
9) All fungi were cultured under the same conditions, but the optimal conditions for growth and degradation of starch could be different for different species. A statement is required to justify this.
We selected standard conditions for the screening of different fungi. We agree with the reviewer that the optimal conditions for growth and degradation of starch could be different in different fungi. Now this is stated in the text.
10) Please shorten this paragraph: The optimal medium for growt of the selected fungi and bacteria was MRS.
Section 3.2 has been rephrased.
11) Could be correct for the selected microbes, but not "usually"
We have corrected the text to state that the selected fungi need high aeration conditions.
12) specify the amount (concentration)
As stated in lines 368-369, the maximum LA production from starch was 72 mM, while from glucose was 200 mM.
13) Simplify figures 2 and 3. Specify number of runs in each case. Add stats.
As suggested also by the other reviewers, we have changed figure 2 to make it more clear, which now includes error bars. Statistical differences were not significant in most cases, but trends are appreciated. All experiments were done in triplicate as described in section 2.4.
14) What is the relevance of biofilm study in this paper? It could be part of another manuscript. If it is essential to this work, then it should be made clear.
As specifies above, we assessed the colonization potential of the microorganisms in a model plastic surface to evaluate the potential to colonize particles of the agro-industrial wastes. This is now stated in section 3.4.
15) Too many information is fed into this manuscript that could form at least two papers. If the authors wish to submit the data in one paper, then they need to : a) reduce/eliminate the content that are not essential (can be taken out without affecting the integrity of the work, and b) simplify the write up for a smoother reading.
As also suggested by the other reviewers, we have tried to simplify the writing and remove unnecessary text.
16) The work done is only at shake-flask level. The mixed-culture behaviour would most likely differ in bioreactors. This fact needs to be specified and elaborated upon (regarding, for example stirring, oxygen availability....
We completely agree with the reviewer, and this is in fact the next experiments and the next publication we are planning.
Reviewer 2 Report
I would like to thank you for nominating this manuscript for evaluation.
The work is good, but I see a number of problems that need to be corrected before the manuscript can be accepted.
1) The title must have the abbreviation PLA in full.
2)The summary corresponds to the general overview of what was done in the research presented, so it must contain mainly the results obtained. In this way, readers will have the dimension of the importance of scientific research and its generated data.
3) Lines 79 to 87: The culture temperature of the different microorganisms, the difference in pH of the culture, different growth times, and productivity are also factors that must be mentioned.
In my point of view, the methodology is problematic since:
a) In general, the authors put many abbreviations without meaning.
b) Line 132: they treat Ascomycetes and lignocellulolytic fungi as separate things (they do not cite the origin of the latter).
c) Line 152: mention the use of lignocellulosic residue for the different combinations, but show only one combination in the results, completely without contextualizing.
d) Line 165: the authors cite a table with several fungi being tested, and suddenly they cite only one fungus being tested in combination, without contextualizing. (Where are the other combinations?)
e) Line 176: Which QSMs????
The results are very messed up.
a) It does not have a statistical analysis.
b) The authors show a table with several fungi and select three, but suddenly they only work with one fungus. However, later the three fungi return.
c) There is no way to know if the results of the LA detection experiments are by HPLC or by another method.
In short: The manuscript needs to undergo a detailed review, and most importantly, organize all the reasoning and support it with statistical analyses.
Author Response
We are really thankful for all the helpful comments from referee 2.
1) The title must have the abbreviation PLA in full.
The abbreviation has been replaced for the full name of PLA in the title.
2) The summary corresponds to the general overview of what was done in the research presented, so it must contain mainly the results obtained. In this way, readers will have the dimension of the importance of scientific research and its generated data.
The abstract has been modified as suggested, including the most relevant results.
3) Lines 79 to 87: The culture temperature of the different microorganisms, the difference in pH of the culture, different growth times, and productivity are also factors that must be mentioned.
A comment has been added into the text in Pag. 2 (lines 91-94).
In my point of view, the methodology is problematic since:
- In general, the authors put many abbreviations without meaning.
We are sorry for overuse of abbreviations. The text has been carefully revised and useless abbreviations have been removed.
- Line 132: they treat Ascomycetes and lignocellulolytic fungi as separate things (they do not cite the origin of the latter).
We agree, it is not clear in the text. We used fungus from our own collection as well as other of public access. We have changed the text in order to clarify it in Pag. 3 (lines 143-144).
- Line 152: mention the use of lignocellulosic residue for the different combinations, but show only one combination in the results, completely without contextualizing.
We have added a short sentence in Pag. 4 (lines 189-190) to contextualize the use of starch from potato residues and beer bagasse in the experiments presented in figure 6 and section 3.5. Preliminary experiments with commercial pure starch from Sigma were made to set the conditions of LA production. “Pure commercial starch” has been added thorough the text to clarify the origin of the polymer in the experiments.
- d) Line 165: the authors cite a table with several fungi being tested, and suddenly they cite only one fungus being tested in combination, without contextualizing. (Where are the other combinations?)
We have added a sentence in Pag. 4 (lines 168-169) to clarify this point.
- e) Line 176: Which QSMs????
We have specified the three QSMs tested in Pag. 4 (line 183).
The results are very messed up.
We apologize. We have rewritten the results according to the referee suggestion.
- a) It does not have a statistical analysis.
In agreement with reviewer 2 we have changed figure 2, which now includes error bars. Statistical differences were not significant in most cases, but trends are appreciated.
- b) The authors show a table with several fungi and select three, but suddenly they only work with one fungus. However, later the three fungi return.
We are sorry for not explaining it conveniently. Only T. amestolkiae was used, as a representative of the 3 fungi, to evaluate the effect of agitation in the consortia and set up the best conditions. The other fungi are expected to behave similarly. Now, this is explained in Pag. 4 (lines 168-169) and in the results section, in Pag. 7 (lines 309-310). However, the presence/absence of QS mechanisms and, in its case, the effect of different QSMs is species-specific, and for this reason we evaluated these phenomena in consortia of the three fungi with the two LAB strains. Finally, the best LA production was achieved in the consortium T. amestolkiae - L. plantarum M9MG6-B2 with dodecanol, and these organisms and QSM were chosen to assay LA production from the agro-industrial wastes.
- c) There is no way to know if the results of the LA detection experiments are by HPLC or by another method.
We have modified the text according to your suggestion, indicating when LA was detected by HPLC or by enzymatic colorimetric kits.
Reviewer 3 Report
The following revisions were proposed prior to publication in this journal:
1. The abstract should be reduced by giving greater emphasis to the results obtained. The initial part could be omitted.
2. Introduction. The methods currently used for the synthesis of lactic acid and the relative disadvantages (e.g. costs, selectivity, etc.) should be more highlighted.
3.The method of presenting the data present in figure 2 appears complex to interpret. Authors are requested to try a different type of graph. Eg dot chart, to better reorganize the acquired data.
4. Very often a limitation of the enzymatic catalysis is associated with the production costs compared to the methods currently in force. Can the authors make an analysis of the costs incurred for the development of this technology?
Author Response
We thank referee 2 for all the helpful comments.
- The abstract should be reduced by giving greater emphasis to the results obtained. The initial part could be omitted.
It has been modified as suggested.
- Introduction. The methods currently used for the synthesis of lactic acid and the relative disadvantages (e.g. costs, selectivity, etc.) should be more highlighted.
This information has been added in Pag. 2 (lines 45-54).
- The method of presenting the data present in figure 2 appears complex to interpret. Authors are requested to try a different type of graph. Eg dot chart, to better reorganize the acquired data.
We have changed the type of graph in Fig. 2 according to the reviewer's suggestion.
- Very often a limitation of the enzymatic catalysis is associated with the production costs compared to the methods currently in force. Can the authors make an analysis of the costs incurred for the development of this technology?
A comment about the economic advantages of our method has been included at the end of the Results section Pag. 14 (lines 583-592).
Round 2
Reviewer 1 Report
I recommend acceptance of the manuscript, but am still with the view that the biofilm study is not relevant to this manuscript. The response of the authors to my comments on the first version is not convincing.
Author Response
We would like to thank Reviewer 1 again for the constructive criticism. The study of the mixed biofilms will be continued in the future to deepen the colonization of the substrate (solid starch residues) and to deepen the association between the fungus and the bacterium.
Reviewer 2 Report
Here are some new suggested fixes!
As mentioned in the first review, it is necessary to add the results to the summary in a more concrete way, with numbers and percentages, to demonstrate the importance of the work.
Minimum adjustments
Line 23: Carbon sources instead C sources
Line 146: Center for Biological Research (CIB) of the Spanish National Research Council (CSIC)
Line 150: De Man, Rogosa, and Sharpe broth (MRS)
Line 160: Czapek Dox agar plates (reference?????)
Line 168: plantarum minimal médium (PMM5)
Line 294: amylase activity is secreted
Line 323: in co-cultures with ??? (some word is missing)
In 3.1 and 3.2 a better discussion of the results is needed
In 3.2 leaving only the matter relating to growing conditions. And in the next section (3.3) talks about LA production.
Figure 2 needs to be corrected, as the name of the microorganisms and their combinations are cut off.
If statistical differences were not significant in most cases is necessary to indicate this in the results.
Line 393 and 394: The authors discuss the results in terms of molarity, but present the results in the figure in terms of percentage (doesn't make much sense).
Where is Fig. S1?
Line 544 to 547: add the reference
Finally, if possible add more current references.
Author Response
We would like to thank Reviewer 2 again for the constructive criticism.
As mentioned in the first review, it is necessary to add the results to the summary in a more concrete way, with numbers and percentages, to demonstrate the importance of the work.
Numerical data has been added to the abstract.
Line 23: Carbon sources instead C sources.
Corrected.
Line 146: Center for Biological Research (CIB) of the Spanish National Research Council (CSIC)
Corrected.
Line 150: De Man, Rogosa, and Sharpe broth (MRS)
Corrected.
Line 160: Czapek Dox agar plates (reference?????)
Added.
Line 168: plantarum minimal médium (PMM5)
Added.
Line 294: amylase activity is secreted
It has been replaced by “extracellular amylase activity secreted”.
Line 323: in co-cultures with ??? (some word is missing)
- amestolkiae has been added.
In 3.1 and 3.2 a better discussion of the results is needed.
We have improved discussion in lines 294-348. We have added text, rephrased some sentences and added some references.
In 3.2 leaving only the matter relating to growing conditions. And in the next section (3.3) talks about LA production.
In 3.2 we have included a preliminary determination of LA production since the goal of this experiment was improving LA concentration. Thus, LA measurement is a parameter that needs to be considered. A sentence has been added to clarify this point. In-depth characterization of LA production is carefully described in 3.3, as suggested.
Figure 2 needs to be corrected, as the name of the microorganisms and their combinations are cut off.
It has been corrected.
If statistical differences were not significant in most cases is necessary to indicate this in the results.
It has been stated in lines 365-366.
Line 393 and 394: The authors discuss the results in terms of molarity, but present the results in the figure in terms of percentage (doesn't make much sense).
We agree. We have also included the values for the percentages obtained in each case.
Where is Fig. S1?
We apologize for this mistake, in the end we decided not to include figure S1 since it does not provide any relevant information for the work. We have now incorporated in L408 that this data is not shown.
Line 544 to 547: add the reference
Added
Finally, if possible add more current references.
Several references have been added.
Reviewer 3 Report
The authors have exhaustively responded to review comments therefore the article is ready for the pubblication
Author Response
We would like to thank Reviewer 3 again for the constructive criticism.